# Reference panel guided topological structure annotation of Hi-C data

Yanlin Zhang[1] & Mathieu Blanchette [1] ✉

Accurately annotating topological structures (e.g., loops and topologically associating domains) from Hi-C data is critical for understanding the role of 3D genome organization in gene regulation. This is a challenging task, especially at high resolution, in part due to the limited sequencing coverage of Hi-C data. Current approaches focus on the analysis of individual Hi-C data sets of interest, without taking advantage of the facts that (i) several hundred Hi-C contact maps are publicly available, and (ii) the vast majority of topological structures are conserved across multiple cell types. Here, we present RefHiC, an attention-based deep learning framework that uses a reference panel of Hi-C datasets to facilitate topological structure annotation from a given study sample. We compare RefHiC against tools that do not use reference samples and find that RefHiC outperforms other programs at both topological associating domain and loop annotation across different cell types, species, and sequencing depths.

Chromosome conformation capture assays such as Hi-C[1], and micro-C[2] have been developed to measure the spatial proximity between DNA fragments in genomes as average pairwise contact frequency in cell populations. These approaches have revealed a hierarchical spatial organization of topological structures of the genome inside nuclei. Among them, topological associating domains (TADs) are kilo- to mega-scale regions with strong interactions between DNA fragments within the same domain and weaker interactions across domains[3]. Loops bring into contact distant loci such as promoters and enhancers[4]. These topological structures are dynamic both within cells[5] and during cellular differentiation[6]. They are essential components of gene regulation.

While Hi-C and its variants remain the most popular approaches to map chromatin contacts on a genome-wide scale, the analysis of the data they produce is challenging, in large part due to the moderate sequencing depth (typically 200–500 Million valid read pairs) compared to the size of the contact frequency matrices that need to be estimated. Numerous TAD annotation tools exist that rely on various statistical significance tests[7–9]. This includes the popular Insulation score (IS)[10], a widely used approach for TAD boundary detection, and more robust variants such as RobusTAD[11]. Still, the performance of all of these approaches is relatively poor, especially at low coverage, due to stochastic noise and biases[7]. Loop detection is even more

challenging[4,12] due to their small size in contact maps. Fit-Hi-C[13] and HiC-DC[14] fit a global model to estimate the background distribution of the contact frequency and identify statistically significant contact pairs by comparing observed values to expected values from the fitted model. These global enrichment approaches evaluate each contact pair independently without modeling neighboring patterns and identify loop clusters instead of discrete loops. In contrast, HiCCUPS[4] compares each contact pair to surrounding regions and identifies locally enriched contact pairs as loops. It requires users to set several sequencing depth sensitive parameters and can only detect loops that satisfy the user defined filtering criteria. Both loop and TAD predictions have been shown to benefit from prior smoothing of Hi-C matrices, e.g., using HIFI[15].

Recent approaches tackle topological structure annotations using computer vision and machine learning techniques. For instance, Mustache[16] treats chromatin loop recognition as a blob-shaped object detection problem. Chromosight[17] employs expert-designed templates to represent each type of topological structures. These generic pattern-based approaches work well on data with sufficient contact pairs but underperform at low sequencing depth. In contrast, Peakachu[12] is a supervised learning approach trained to recognize loops using data from orthogonal experiments as target values.

[1]School of Computer Science, McGill University, Montréal, Québec H3A 0E9, Canada. ✉e-mail: blanchem@cs.mcgill.ca

Many approaches have been introduced to address the issue of insufficient sequencing depth. Grinch[18] proposed a graph-regularized non-negative matrix factorization algorithm to smooth sparse Hi-C contact map while detecting TADs. DeepLoop[19] identifies significant interactions from sparse Hi-C contact maps by denoising and enhancing loop signals with a neural network. Higashi[20], a single-cell Hi-C data analysis tool, represents a cohort of scHi-C data as a hypergraph, learns to predict missing hyperedge to impute missing interaction, and then performs structural annotation on imputation.

A common strategy in analyzing biological data is to complement data about the sample of interest with data of the same type obtained previously for other samples. This strategy has proven effective for genotype imputation[21] and phasing[22], as well as protein structure prediction[23], among others. Even though hundreds of Hi-C experiments have been conducted, they have never been analyzed jointly for topological structure annotation. Here we introduce RefHiC, a reference panel informed deep learning approach for topological structure (loop and TAD) annotation from Hi-C data. RefHiC uses a reference panel that contains high-quality Hi-C data of different cell types. For each potential contact in the study sample, it uses an attention mechanism[24] that determines which of the reference samples are most relevant, and then makes a prediction based on the combined study sample and attention-weighted reference samples. We demonstrate that RefHiC enables significant accuracy and robustness gains, across cell types, species, and coverage levels.

## Results

### Overview of RefHiC

RefHiC takes as input a Hi-C contact map for a study sample and a reference panel of Hi-C contact maps (provided with the tool). It produces highly reliable loop or TAD boundary annotations for the study sample. RefHiC is based on two components (Fig. 1 and "Methods"): (i) a neural network predicts loop (resp. TAD boundary) scores for every candidate pair (resp. locus) based on the local contact sub-matrix, combining information from the study sample and the reference panel; (ii) a task-specific component selects one representative loop/TAD boundary from each high-scoring cluster. For human, the reference panel contains 30 uniformly processed Hi-C contact maps, each with at least 350 million contact pairs (Supplementary Table 1). For mouse, it consists of 20 such maps (Supplementary Table 2). Normalization of reference Hi-C samples is unnecessary as the network automatically learns to handle batch effect and coverage differences from the training data.

To obtain a loop or TAD boundary score for bin pair $(i, j)$ (with $i \neq j$ for loops and $i = j$ for TAD boundaries), an encoder projects the sub-matrices centered at $(i, j)$ in both the study sample and reference panel to low-dimensional embeddings. An attention module[24] computes a combined representation of all reference samples as a weighted sum of their embeddings, with weights based on their local similarity to the study sample's embedding. Finally, a multi-layer perceptron predictor computes loop or TAD boundary score from the concatenation of the study sample's embedding and the attention output. The process is repeated

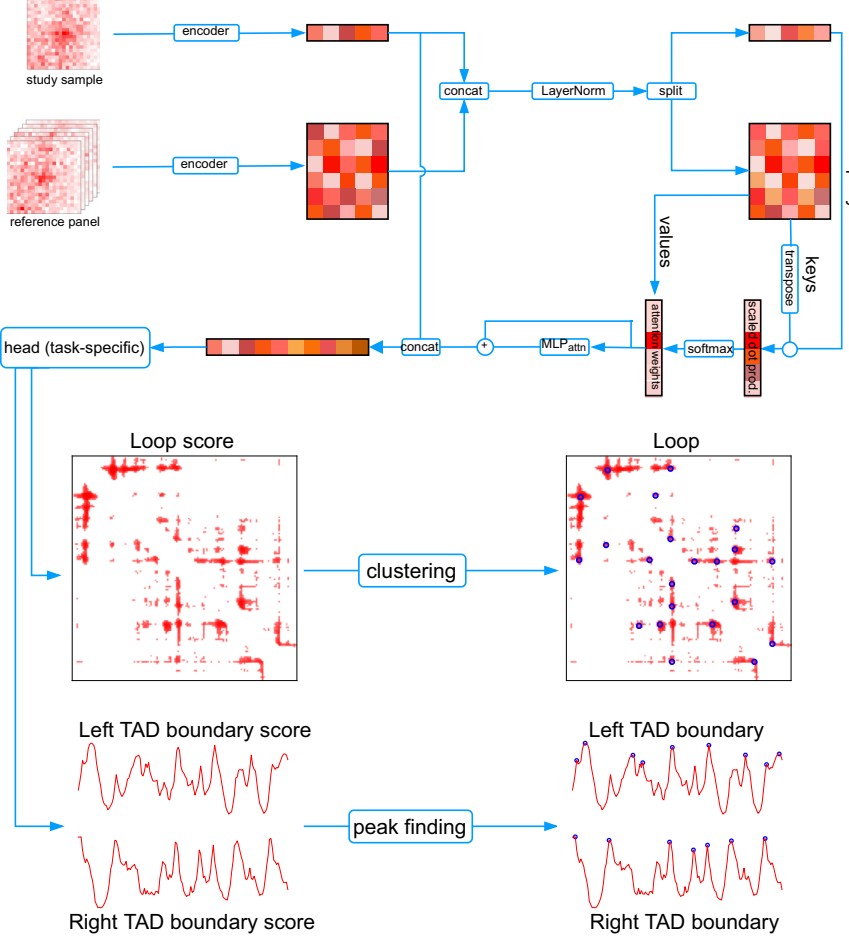

**Fig. 1 | RefHiC architecture.** Overview of the RefHiC neural network for loop and TAD boundary scoring, followed by clustering or peak finding algorithm for discrete loop and TAD predictions (shown as blue circles).

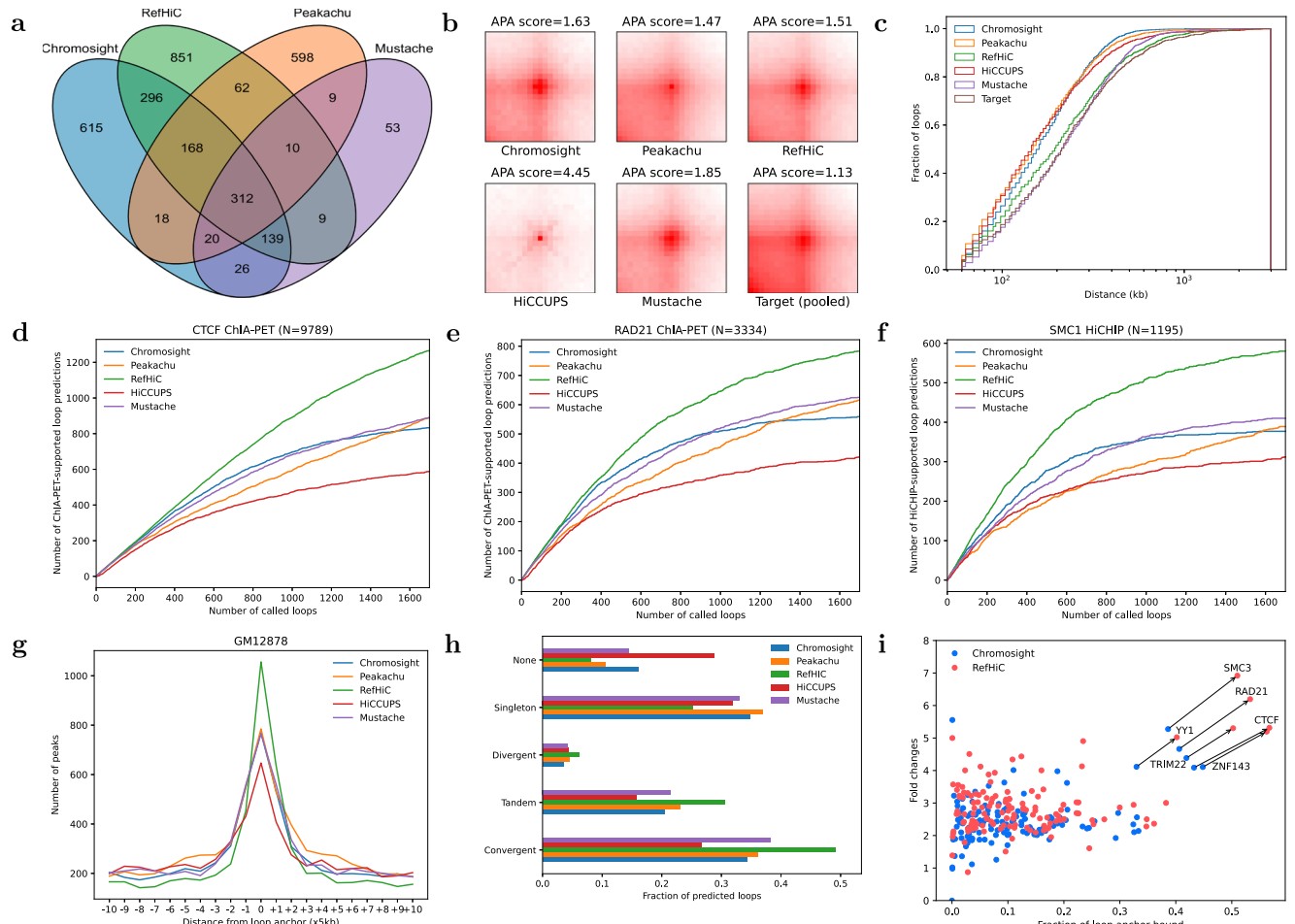

**Fig. 2 | Comparison of RefHiC, Chromosight, Peakachu, HiCCUPS, and Mustache on GM12878 Hi-C data (500M valid read pairs). a** Venn diagram of loops predicted by different tools. **b** Aggregate peak analysis profiles for target (ChIA-PET and HiCHIP identified) and annotated loops. **c** Cumulative distance distributions of predicted loops. RefHiC's predicted loop distance distribution closely resembles that of ChIA-PET/HiCHIP-supported loops (target). **d–f** Number of ChIA-PET/HiCHIP-supported loop predictions, among the top 1700 predictions made by RefHiC and other tools, for test chromosomes chr15, chr16, and chr17, compared against CTCF ChIA-PET (**d**), RAD21 ChIA-PET (**e**), and SMC1 HiCHIP (**f**). RefHiC's loop predictions matches those experimental data better than predictions made by other tools on test chromosomes. **g** Occupancy of ChIP-seq identified CTCF binding site as a function of distance to predicted loop anchors. **h** Orientation of CTCF motifs at predicted loops. **i** Transcription factor (TF) occupancy at predicted loops (RefHiC and Chromosight only). Each dot is a TF or histone modification (based on 133 ENCODE ChIP-seq data sets for GM12878), whose x-coordinate is the fraction of loop anchors containing a binding/modification site and the y-axis is the fold enrichment against genome-wide frequency. Most TFs are more strongly enriched at RefHiC loop predictions than at Chromosight loop predictions.

for every pair $(i,j)$ to obtain a scoring matrix (resp. vector), from which discrete loop (resp. TAD boundary) predictions are extracted.

Training RefHiC (i.e., choosing the weights of the encoder, fully connected layers in attention module, and head) is achieved using a variety of downsampled versions of a high-coverage Hi-C data set for GM12878[4]. Following Salameh et al.[12], we used as prediction targets a set of long-range loops identified at 5 kb resolution by either ChIA-PET on CTCF[25] or RAD21[26], as well as by HiCHIP on SMC1[27] or H3K27ac[28]. Using multiple experimental data sets ensures a broad coverage of various types of loops.

Importantly, although RefHiC is trained on GM12878 data, the model learned is not cell-type specific, and we will demonstrate in later sections that the same model can be used to annotate structures in many other cell types without retraining and with similar accuracy. The same trained model can also be used to make predictions on mouse Hi-C data, based on our reference panel for that species.

In our experiments, we used human chromosomes 11 and 12 for validation, chromosomes 15–17 for testing, and the rest of the autosomes for training. To prevent potential data leakage, all results reported here pertain only to the three test chromosomes.

## RefHiC accurately detects chromatin loops from Hi-C contact maps

We first assessed the loop prediction accuracy of RefHiC on a downsampled Hi-C data set (500M valid read pairs) for human GM12878 cells[4]. We then applied Chromosight[17], Peakachu[12], Mustache[16], and HiCCUPS[4] to annotate loops from the same data with default parameters. For all tools, we set the same 5% FDR cutoff whenever possible.

The sets of predicted loops are quite different among tools, with RefHiC making the largest number of unique predictions Fig. 2a and Supplementary Fig. 1. Aggregate peak analysis (Fig. 2b) shows that loops detected by Chromosight, RefHiC, and Mustache had a more diffuse loop center compared to those identified by Peakachu and HiCCUPS. Finally, the distribution of distances between loop anchors predicted by RefHiC and Mustache most closely resembled that of ChIA-PET/HiCHIP-supported loops (Fig. 2c), whereas Peakachu, HiC-CUPS, and Chromosight predicted more short-range interactions.

We then evaluated predicted loops by comparing them to loops revealed by loop-targeting experimental data, allowing up to 5 kb shift. To facilitate interpretation, we considered the top 1700 predictions from each tool by adjusting the FDR or loop score cutoff. Figure 2d–f

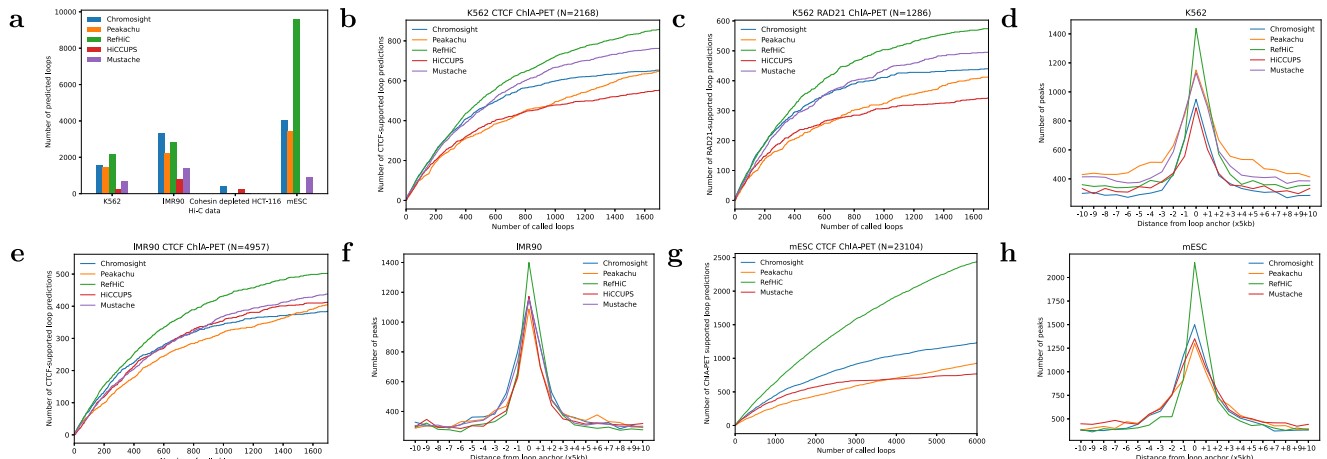

**Fig. 3 | Loop detection in Hi-C data from human K562, IMR90, and cohesin-depleted HCT-116 cells, as well as mouse ESC. a** Number of loops identified in Hi-C datasets obtained in each data set (human data: test chromosomes 15–17 only; mouse data: all autosomes). Note that in HCT-116, one would not expect any cohesin-mediated loops. **b**, **c** Number of ChIA-PET/HiCHIP-supported loop predictions, among the top predictions made by RefHiC and other tools on K562 Hi-C data, for test chromosomes chr15-17, compared against CTCF (**b**) and RAD21 ChIA-PET (**c**) data. **d** Occupancy of ChIP-seq identified CTCF binding sites in K562 cells as a function of distance to predicted loop anchors. **e**, **f** Same as (**b**, **d**), but for data obtained in IMR90 cells. **g**, **h** Same as (**b**, **d**), but for data obtained in mESC (all autosomes).

and Supplementary Fig. 2a show that RefHiC produced 1250 CTCF-supported loops, 784 RAD21-supported loops, 588 SMC1-supported loops, and 213 H3K27ac-supported loops. In contrast, other tools yielded 20–52% fewer validated loops. Comparison against DeepLoop[19] reveal similar numbers (Supplementary Note 1 and Supplementary Fig. 3). Finally, to delineate the impact of using a reference panel, we evaluated a version of RefHiC that operates exclusively based on the study sample (Supplementary Note 2); while this reference-free predictor obtains state-of-the-art performance (or better), it is far from the reference panel based RefHiC (Supplementary Fig. 4). As shown in Fig. 2g, predicted loop anchors detected by RefHiC were strongly enriched with the CTCF binding motifs. TAD-forming loops have been previously shown to be associated with the presence of convergent CTCF binding sites at loop anchors[5]. Indeed, 50% of RefHiC's loop predictions are associated with such pairs of sites; significantly more than for other tools (Fig. 2h).

Among loops detected by each tool, 46% of RefHiC, 39% of Chromosight, 50% of Peakachu, and 9% of Mustache were not detected by other tools (Fig. 2a). Supplementary Fig. 5 shows that RefHiC-specific predictions are not only more numerous but also more accurate when evaluated against CTCF/RAD21 ChIA-PET, and SMC1 HiCHIP data, though slightly less accurate than Chromosight and Peakachu on H3K27ac HiCHiP data. Chromosight and Peakachu were slightly better than RefHiC when being evaluated against H3K27ac HiCHiP data. A deeper analysis of the properties of loops predicted by each tool is presented in Supplementary Note 3 and Supplementary Figs. 6–8.

To further study the properties of loops predicted by each tool, we performed transcription factor (TF) and histone modification enrichment analysis around loop anchors. Figure 2i and Supplementary Fig. 2b, c show enrichment for known loop-mediating proteins (SMC3, RAD21, YY1, TRIM22, CTCF, and ZNF143) was strongest for RefHiC compared to Chromosight and Peakachu, and comparable to Mustache.

Combined, these results demonstrate the overall superior prediction accuracy of RefHiC on GM12878 data (500M read pairs) compared to other approaches.

## RefHiC performs well across cell types and species

Although RefHiC is trained on human GM12878 data, we demonstrate here that the same trained model performs well across other human and mouse cell types. We applied RefHiC and other tools (5% FDR) to Hi-C data from human K562, IMR90[4], and cohesin-depleted HCT-116[29]

cell lines (test chromosomes 15–17 only), as well as mouse embryonic stem cells (mESC)[30] (all chromosomes). Since the IMR90 data set has twice the sequencing coverage of the K562 data set, all tools identified more loops in the former, with Chromosight and RefHiC making the largest number of predictions (Fig. 3a). However, RefHiC is notably more robust to sequencing depth, with a decrease of only 22% from IMR90 to K562, compared to 34–66% for other tools.

Cohesin-depleted HCT-116 cells are expected not to contain any loop. Indeed, RefHiC, Peakachu, and Mustache made fewer than 24 loop predictions on this data, whereas Chromosight and HiCCUPS had many more likely false positives.

For the mESC data, which contains only 124M valid read pairs, we used the same RefHiC model trained from human GM12878, but with a mouse reference panel made of 20 mouse Hi-C data sets (Supplementary Table 2). Applied to the complete set of autosomes, RefHiC identified more than twice as many loops as any other tool, indicating that it is much more sensitive than other tools on low-coverage data.

We then assessed these tools' accuracy using loops revealed by orthogonal experiments. As before, we included the top 1700 predictions on test chromosomes from each tool by adjusting FDR or loop score cutoff. For K562 data, as shown in Fig. 3b, c, RefHiC outperformed other tools as it identified more CTCF- and RAD21-supported loops. The pileup analysis of CTCF binding sites around predicted loop anchors (Fig. 3d) shows occupancy 25% higher for RefHiC than for Peakachu and Mustache, and 51% higher than for Chromosight and HiCCUPS. Similar results are obtained on IMR90 data (Fig. 3e, f), although its very high coverage enables competing approaches to get somewhat closer to RefHiC's performance. For the mESC data, Fig. 3g, h indicates that RefHiC outperformed alternative tools significantly as it detected as twice as many CTCF-supported loops as alternative tools and its loop anchor predictions are more strongly enriched for CTCF binding sites than other tools. To further study the ability of RefHiC to identify loops in samples that are very different from those present in its reference panel, we generated reference panels excluding samples that are closely related to study sample GM12878, or even entirely unrelated (e.g., from the incorrect chromosome). Supplementary Note 4 and Supplementary Fig. 14 show that RefHiC performs comparably or better than other tools even under this less favorable scenario. Together, the results show that RefHiC achieves superior performance across both human and mouse cell types.

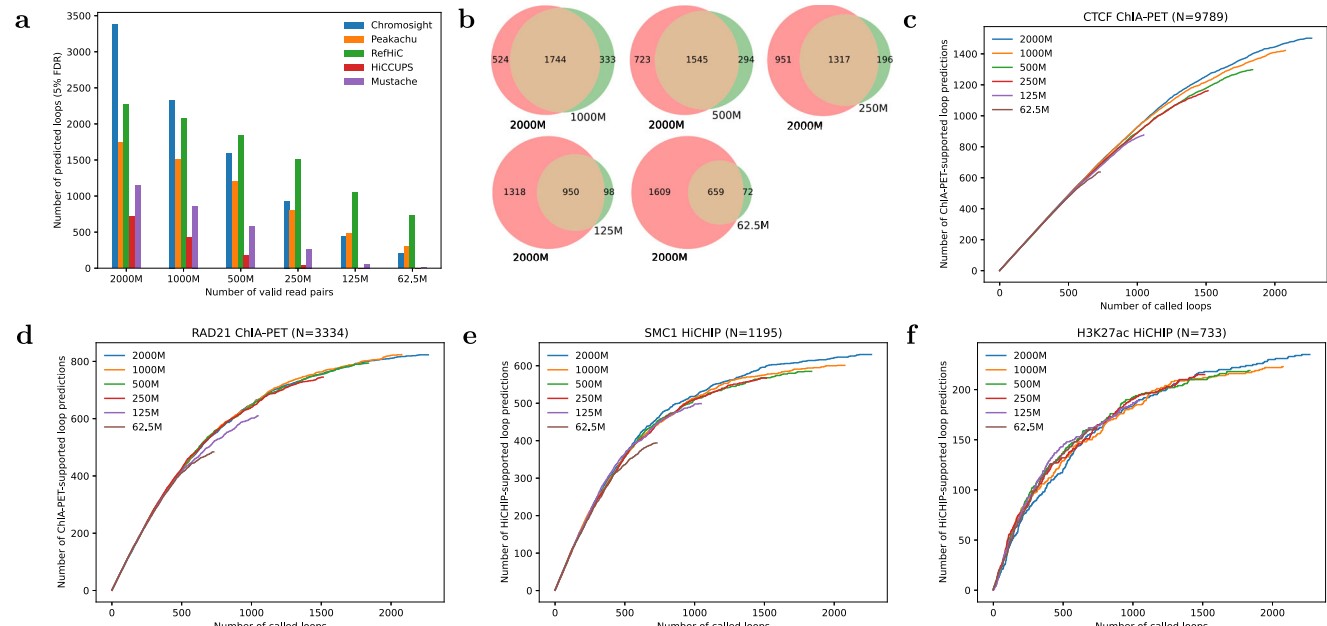

**Fig. 4 | Detection of loops at lower sequencing depths. a** Number of loops predicted by different tools at 5% FDR, for decreasing number of valid intra-chromosomal read pairs. **b** Venn diagram of loops predicted from Hi-C data of different sequencing depths by RefHiC. **c–f** Number of RefHiC loop predictions supported by experimental GM12878 ChIA-PET/HiCHIP data (test chromosomes chr15-17) at different levels of sequencing coverage: CTCF ChIA-PET (**c**), RAD21 ChIA-PET (**d**), SMC1 HiCHIP (**e**), H3K27ac HiCHIP (**f**).

## RefHiC is robust to sequencing depths

To benchmark RefHiC's ability to detect loops from Hi-C data at different sequencing depths, we produced downsampled versions a high-coverage GM12878 Hi-C combined contact map[4] and applied different loop prediction tools (default parameters; FDR cutoff 0.05 when possible). Although lower sequencing depths led to fewer loop predictions for all tools (Fig. 4a), RefHiC was most robust to sequencing depths. For example, RefHiC identified 731 loops from low coverage Hi-C data (62.5M contact pairs)−32% of the results obtained from 2000M contact pairs. In contrast, other tools are largely unable to make sensitive loop predictions at this low sequence depth. Figure 4b shows that RefHiC detected highly concordant sets of loops across sequencing depths: ~85% of loops annotated from Hi-C data containing 1000M, 500M, and 250M contact pairs overlapped those annotated from the 2000M contact pairs data set. This percentage was even higher (90%) on low-depth Hi-C data (i.e., 125M, and 62.5M contact pairs). This shows that not only is RefHiC capable of detecting a good number of loops in low-coverage data, but it also does not introduce significantly more false positives. Figure 4c−f confirm that RefHiC predictions on low-depth data sets maintain a very high level of accuracy when evaluated against loops mediated by CTCF, RAD21, SMC1, and H3K27ac. In short, this means that predictions made on low-coverage data are nearly as specific as those made on the full data, but are simply less sensitive. At all sequencing depths, RefHiC achieved higher accuracy than alternative tools (Supplementary Fig. 4). This superior robustness, accuracy, and reliability is attributable to the use of reference panel.

## RefHiC identifies both rare and common loops

Since RefHiC uses a reference panel to complement data from the study sample, one may expect that it performs best on common loops (i.e., those present in a large number of cell types from our reference panel). To determine the prevalence of each loop, we ran Mustache and Chromosight on our reference samples and merged their predictions (allowing a 2-bin shift; the two tools failed on 10 of the 29 samples as one or both detected less than 10 loops, leaving a total of 19 samples

annotated). We then assessed the frequency at which loops predicted by RefHiC on GM12878 were found in the 19 reference samples. The distribution of reference panel frequencies among loops predicted by RefHiC resembled that of Peakachu, Mustache, and Chromosight (Fig. 5). For all these tools, the majority of highest-scoring loops were found to be present across nearly all samples, suggesting that constitutive, non-cell-type specific loops have features that make them easily predictable. Still, more than 20% of loops predicted by RefHiC are rare (found in at most 5 of the panel data sets), and 4% are specific to GM12878, demonstrating that the use of a reference panel does not strongly bias the results in favor of common loops. Still, those proportions are slightly lower than those obtained with the three other tools, which could be explained by a combination of a weak bias toward common loops for RefHiC, and an increased false-positive rate (which usually will appear as cell-type specific loops) for the other tools. Indeed, the number of GM12878-specific loop predictions that are supported by experimental data is actually comparable across tools (Supplementary Fig. 9). Peakachu identified more cell-type specific validated loops than other tools, but with a lower specificity than RefHiC. Among loops found to occur at least once in the panel, RefHiC gets more ChIA-PET/HiCHIP-supported predictions than alternative tools (Supplementary Fig. 9e, g−t), except that Peakachu identified more H3K27ac HiCHIP-supported loops (Supplementary Fig. 9f). Finally, loops predicted by HiCCUPS were very different, containing more of what looks like GM12878-specific loops (i.e., loops absent from the reference panel), many of which are likely false-positives.

## RefHiC accurately detects TADs

RefHiC is a versatile framework for topological structure annotation. Here we show that RefHiC can detect TADs once trained using as target values RobusTAD TAD boundary scores obtained on a high-coverage HiC data set (see "Methods"). We first compared RefHiC's performance on downsampled versions of a GM12878 Hi-C contact map to that of two established TAD boundary predictors (RobusTAD[11] and insulation score[10]). Figure 6a, b and Supplementary Fig. 15 show that at 500M valid read pairs, RefHiC and RobusTAD succeed at identifying a similar

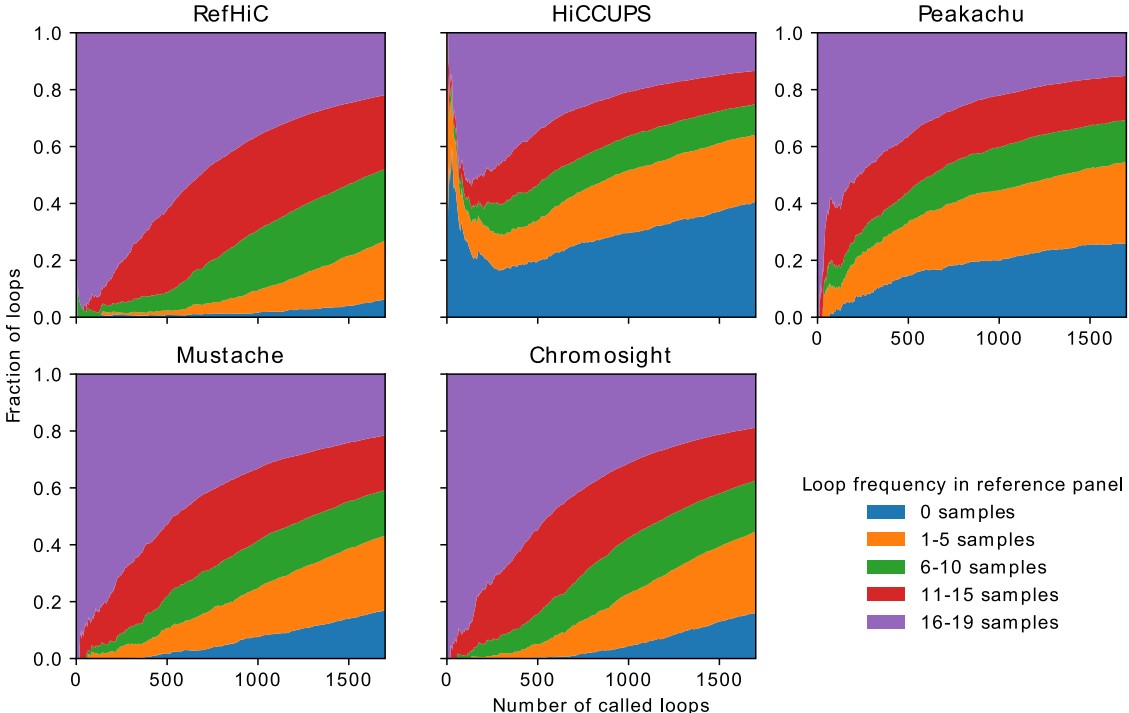

**Fig. 5 | Comparison of tools' ability to identify rare and common loops.** Loop frequency within the reference panel is assessed based on Mustache and Chromosight's predictions on individual reference Hi-C data sets.

total number of CTCF-supported TAD boundaries, although RefHiC's specificity is much higher considering that its total number of predictions (at 5% FDR) is approximately 40% less than with RobusTAD. Supplementary Figs. 10 and 11 show that at very low coverage (125M and 62.5M valid read pairs), RefHiC achieves both higher sensitivity and specificity. In all cases, both RefHiC and RobusTAD outperform Insulation score.

We then benchmarked RefHiC against 13 TAD callers (TopDom[31], Armatus[32], deDoc[33], Arrowhead[4], HiTAD[34], EAST[35], OnTAD[36], CaTCH[37], Grinch[18], Domaincall[38], GMAP[39], HiCSeg[40], and IC-Finder[41]) on test chromosomes 15–17. Because there is no universally accepted gold-standard TAD annotation to compared against, we evaluated various aspects of the predictions made by the different tools. We first compared the number and size of TADs identified by each tool (Fig. 6c, d). Although the number varies from 347 to 3499, most tools (including RefHiC) identified 1000–1500 TADs, with median TAD size around 130 kb for RefHiC. TADs are domains with high levels of internal interaction, so one measure of TAD annotation quality is the average observed/expected ratio within TADs (Fig. 6e). RefHiC's TAD predictions are among the densest in interaction frequencies. We then calculated the enrichment for ChIP-Seq signals of structural proteins known to be associated with TAD boundaries (i.e., CTCF, RAD21, and SMC3)[8] at predicted TAD boundaries and nearby (Fig. 6f and Supplementary Fig. 16). Based on this metric, RefHiC is only outperformed by Arrowhead, which identifies 3 times fewer TADs. Histone marks usually correlate with regulatory activity, and most TADs are typically enriched for either activation (H3K36me3) or repression (H3K27me3) marks, but rarely both. We calculated the ratio between H3K27me3 and H3K36me3 within each TAD prediction and counted the fraction of TAD predictions where this ratio was particularly large or small (see "Methods"). RefHiC is among the top three TAD callers under this metric (Fig. 6g), only bested by tools that predict a much smaller number of TADs (Arrowhead and CaTCH). Many TADs in mammalian genomes exhibit a strong contact between their left and right boundary loci, forming a visible TAD corner; they are often referred to as loop domains. We compared predicted TAD corners against CTCF

ChIA-PET data (Fig. 6c). RefHiC is the best-performing tool, with 556 (36.5%) TADs corners supported by CTCF ChIA-PET data (allowing 1-bin mismatch). Finally, we evaluated the prediction reproducibility at both the boundary and full TAD levels when TAD callers are applied to Hi-C data containing different numbers of valid read pairs. RefHiC proved much more robust than other tools at the TAD boundary prediction task (Fig. 6h) and better than most (but slightly worse than GMAP and HiCSeg) at the full TAD prediction task (Fig. 6i). This last observation is likely is due to the fact that pairing predicted TAD boundaries to obtain full TAD predictions is a step that does not currently take advantage of RefHiC's reference panel.

## Discussion

Here we present RefHiC, a deep learning framework that utilizes a reference panel to guide the annotation of topological structure from a given study sample. In contrast, existing topological structure detection algorithms are study-sample based (i.e., reference-free) detectors and hence their ability to reliably detect topological structures from typical sequencing depth Hi-C data is limited. Our extensive evaluation demonstrated that RefHiC outperforms existing tools for both TAD and loop annotations, in data sets ranging from very high to very low sequencing coverage, with the most striking improvements observed in the latter case. This benefit comes at little cost in terms of RefHiC's ability to identify cell-type specific loops.

Importantly, although RefHiC is a machine-learning-based model trained primarily on GM12878 Hi-C data, the same trained model is effective on different cell types, at different levels of coverage, and across human and mouse. Indeed, all results reported here for loop prediction were obtained with the same trained model, which is available in our GitHub repository. This model can be used for mammalian Hi-C data analyses without retraining. Retraining would only be needed if other types of structures are sought, or if the experimental protocol used to generate the study sampled differed significantly from the standard in situ Hi-C protocol. In such cases, RefHiC would require retraining but would still be able to take advantage of our Hi-C reference panel, i.e., the reference panel does

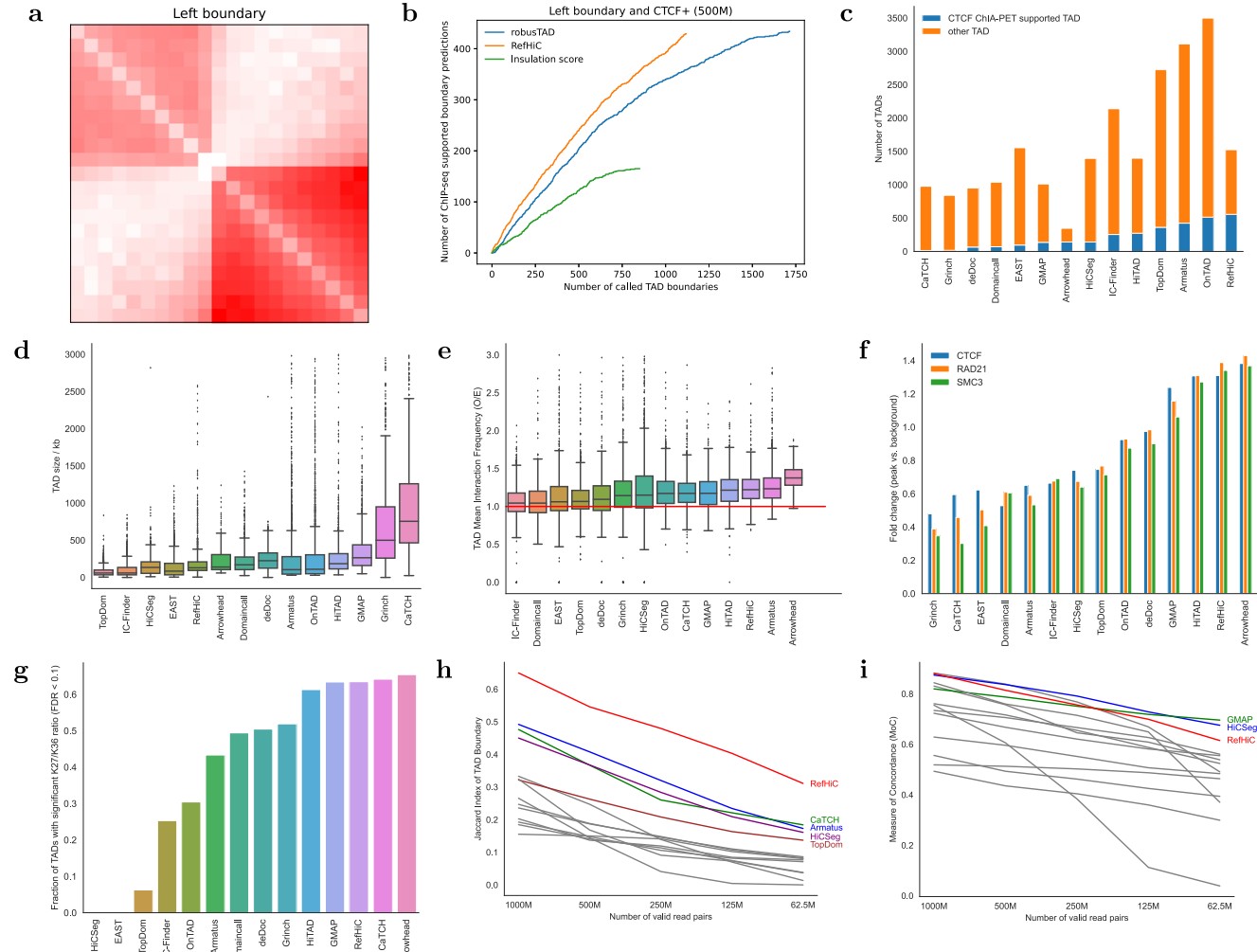

**Fig. 6 | Detection of TAD boundaries and TADs on GM12878 Hi-C data. a** TAD boundary pileups for left boundaries predicted by RefHiC. **b** Number of predicted left TAD boundaries supported by ChIP-seq identified CTCF binding sites (positive strand only), for RefHiC, RobusTAD, and Insulation score. **c–i** Benchmarking RefHiC against 13 other TAD callers on TAD annotation. **c** Number of TADs predicted by different tools, and proportion of predicted TAD boundary pairs that are supported by CTCF ChIA-PET data. Size (**d**), and mean interaction frequency (observed/expected) (**e**) of TAD predictions. The number of TADs used to generated box plots are provided in (**c**). In each box, the upper edge, central line, and lower edge represent the 75th, 50th, and 25th percentile, respectively. Upper whiskers represent 75th percentile + 1.5 × interquartile range (IQR), lower whiskers represent minimum values, and dots represent samples above the 75th percentile + 1.5 × IQR. **f** Enrichment of CTCF, RAD21, and SMC3 peak signals at TAD boundaries. **g** Fraction of TADs predicted by each caller with a significant (high or low) H3K27me3/H3K36me3 log10-ratio (FDR < 0.1). Jaccard index of predicted TAD boundaries (**h**) and Concordance between TADs (**i**) predicted on high-coverage GM12878 data (2B valid read pairs) compared to those predicted on downsampled Hi-C data. Note: **a–g** are based on a downsampled GM12878 Hi-C data set that containing 500M valid read pairs).

not need to be of the same type as the study sample. However, applying RefHiC to Hi-C data obtained from other species might be more challenging, due to the lacking of reference samples, than reference-free alternative tools.

Our method has several methodological contributions. The key innovation of RefHiC is the introduction of a Hi-C reference panel. Our attention-based framework enables RefHiC to identify and take advantage of the reference samples that exhibit similar local structures as the study sample at the locus pair of interest. This approach based on local similarity significantly outperformed an analogous approach based on global similarity (Supplementary Note 5). Besides, we introduced contrastive pretraining[42] and data augmentation by downsampling Hi-C contact map techniques to train a single model capable of handling Hi-C data of different sequencing depths. We believe this training procedure can improve many machine learning applications for Hi-C data analysis[12,43–45].

In principle, reference-based approaches such as RefHiC have the potential of becoming increasingly accurate as larger compendia of high-quality Hi-C data sets obtained from diverse cell types become available and get included in the panel. However, our analyses (Supplementary Fig. 12) suggest that limited benefits for the analysis of GM12878 data are obtained beyond a panel consisting of 10 high-coverage data sets. However, we expect that this observation is dependent on the origin of the study sample of interest, and RefHiC's performance on study samples that are divergent from the cell types represented in the panel would certainly benefit from additional, closer reference samples.

RefHiC could potentially be improved in several directions. Expanding the reference panel could improve prediction accuracy, but this is challenging memory-wise with our current implementation. In addition to further software optimization, we will develop high-diversity panels that will aim to capture most of the structural diversity through a moderate number of Hi-C data sets. In addition, we can potentially extend RefHiC to analyze data at an even higher resolution (e.g., 1 kb), although this too would require optimizing data handling to limit the memory footprint and IO time.

Across the different sub-fields of data-driven biology, major leaps forward have taken place when researchers have developed approaches that enabled the analysis of one data set to benefit from the availability of other published data sets. RefHiC is an approach to enable this type of reference-panel-based analysis of 3D genomics data. It enables high-accuracy annotation of Hi-C data sets even at moderate sequencing coverage, and boosts the accuracy of the analysis of even the most deeply sequenced data sets. RefHiC and other approaches of its kind have the potential to become an essential method for topological structure annotation from Hi-C contact maps, paving the way to further our understanding of 3D genome organization and functional implications. With the increasing availability of high-quality Hi-C data sets from diverse cell types, we anticipate that the power of RefHiC will further develop.

## Methods

### RefHiC model architecture

The RefHiC network consists of three parts (see Fig. 1 and Supplementary Fig. 13): (i) an encoder, (ii) an attention module, and (iii) a task-specific head. The encoder takes an input of dimension $(2 \times w + 1) \times (2 \times w + 1) \times 2$, where $w$ is the window size ($w = 10$ in loop annotation, $w = 20$ in TAD boundary annotation) and projects the input to a $d$-dimensional embedding ($d = 64$). It is built with one ReLU-activated convolution layer with kernel size three and two ReLU-activated fully connected layers with $d$ hidden units in each layer. In forward pass, the encoder computes an embedding $\mathbf{e_s} \in \mathbb{R}^{1 \times d}$ for the study sample and $[\mathbf{e_1}, \mathbf{e_2}, \ldots, \mathbf{e_n}] \in \mathbb{R}^{n \times d}$ for the $n$ reference samples. The attention module takes as input the embeddings for both the study and reference samples and outputs $\mathbf{a} \in \mathbb{R}^{1 \times d}$ that contains topological structural information learned from the reference panel. The layer-normalized study sample's embedding is used as query ($\mathbf{Q} \in \mathbb{R}^{1 \times d}$) against the layer-normalized reference samples' embeddings, which are used as both keys ($\mathbf{K} \in \mathbb{R}^{n \times d}$) and values ($\mathbf{V} \in \mathbb{R}^{n \times d}$). We define the attention weights $\boldsymbol{\alpha} = \mathrm{softmax}(\mathbf{Q}\mathbf{K}^T) \in \mathbb{R}^{1 \times n}$, where $\alpha_j$ represents the relative amount of attention paid to sample $j$ in our reference panel when analyzing the study sample. The attention output $\mathbf{a}$ is computed as,

$$\mathbf{a} = \mathrm{softmax}\left(\mathbf{Q}\mathbf{K}^T\right)\mathbf{V} + \mathrm{MLP_{attn}}(\mathrm{softmax}\left(\mathbf{Q}\mathbf{K}^T\right)\mathbf{V}) \qquad (1)$$

where $\mathrm{MLP_{attn}}$ has ReLU-activated fully connected layers with two hidden layers, and each layer contains $d$ hidden units. Finally, the head is a task-specific predictor (either for loop or for TAD boundary prediction) with 2 hidden layers containing $2d$ and $d$ hidden units. It has one sigmoid-activated output unit for loop prediction and two tanh-activated output units for TAD boundary prediction. Both tasks use the concatenation of the study sample's embedding $\mathbf{e_s}$ and attention output $\mathbf{a}$ as input. For loop prediction, it outputs a value indicating loop probability. For TAD boundary prediction, it outputs two values corresponding to left and right boundary scores. To make predictions, we apply RefHiC to each entry in the upper triangular contact matrix to compute loop probabilities, and each entry on the main diagonal to compute TAD boundary scores.

### Detecting loops by density-based clustering

Applied to the window centered around each bin pair $(i, j)$ (a.k.a pixel), RefHiC produces a loop probability score $L(i,j)$. Pixels where $L(i,j) > 0.5$ are called *loop candidates*. Candidate $(i,j)$ is called an *isolated* prediction if there are less than six candidates within a 5-bin by 5-bin square centered at $(i,j)$. We excluded all isolated predictions as they are likely to be false positives. We then grouped the remaining candidates into clusters using a density-based clustering algorithm[46]. We first computed local density $\rho(i,j)$ for candidate $(i,j)$ by convolving scores with a Gaussian kernel over candidates $(i',j')$ where $\min\{|i'-i|,|j'-j|\} \leq 5$. We

then calculated $\delta(i,j)$ as the minimum Chebyshev distance between candidate $(i,j)$ and any candidate $(i',j')$ with higher density. For candidates $(i,j)$ with the highest local density, we defined it as $\delta(i,j) = \delta_{\max}$, where $\delta_{\max}$ is a large constant. We used a KD-tree data structure to facilitate the fast computation of $\delta(i,j)$. We discarded candidates with $\delta$ smaller than five since they were more likely to be redundant annotations. Among the remaining candidates, we then used a target-decoy search approach to find cluster centroids by identifying candidates with high local density. Given a study sample Hi-C contact map, we created a decoy contact map by permuting interaction frequencies diagonal-wise, applied RefHiC to detect loop candidates in the decoy contact map, and calculated $\rho$ and $\delta$ for loop candidates in the decoy contact map. We then sorted candidates predicted from the study and decoy samples based on local density ($\rho$) and selected the top candidates while keeping the false discovery rate (FDR) at $\alpha = 0.05$. Last, we assigned the remaining candidates to their nearest clusters and chose as a loop the highest local density candidate in each cluster.

### Detecting TAD boundary by peak finding

RefHiC annotates right and left TAD boundaries separately. To annotate discrete right boundaries, we represented right boundary scores produced by RefHiC as sequential data and annotated boundaries by finding peaks using the find_peak function in SciPy[47]. When selecting TADs, we used the target-decoy search approach to find the height (i.e., score cutoff) parameter in find_peak. We also set the minimum distance between peaks to 5 to exclude locally redundant TAD boundaries. We applied the same steps to annotate left boundaries from left boundary scores. Like TopDom and GMAP, we annotate a region starting from a left boundary $l_i$ and ending at a downstream right boundary $r_j$ ($r_j$ is on the left of or identical to $l_{i+1}$) as a TAD. We allow a left boundary pairs with multiple right boundaries. This produces nested TADs.

### Feature vector and training data

RefHiC's feature vector is defined as a tensor with two channels (observed interaction frequency and observed/expected ratio) in the shape of $2 \times (2 \times w + 1) \times (2 \times w + 1)$, corresponding to the window of size $2w + 1$ centered at the pixel of interest. $w$ is a hyperparameter set to $w = 10$ for loop annotation and $w = 20$ for TAD boundary annotation at 5 kb resolution. We trained RefHiC with Hi-C contact maps downsampled from the combined GM12878 Hi-C contact map[4].

For loop annotation, following Salameh et al.[12], we used as gold-standard (i.e., positive training cases) a set of long-range loops identified by either ChIA-PET on CTCF[25] or RAD21[26], as well as by HiCHIP on SMC1[27] or H3K27ac[28]. Using multiple experimental data sets ensures a broad coverage of various types of loops. We binned interactions at 5 kb resolution and removed duplicates and any contact pairs with a distance shorter than 50 kb or longer than 3 Mb, resulting in 74,855 interactions used as positive cases for loop annotation. We created the negative set by selecting non-loop pairs of different types: (i) We randomly drew 50,000 contact pairs, excluding contact pairs with Chromosight scores greater than 0, while preserving the distance distribution between positive loop anchors, (ii) to increase the representation of long range negative examples, we randomly selected 10,000 long range (1 - 3 Mb) pairs, most cases of (i) and (ii) are non-loop pairs in all samples. The negative set does not contain enough data representing pairs of loci that do not form a loop in the study sample, but do in some of the reference samples. Thus, we select examples (iii) from pairs identified as loops in one or more reference samples: we applied Chromosight and Mustache with default parameters on all samples in the reference panel to annotate loops, merging annotations while excluding duplicates (allowing 1-bin mismatch). Last, we merged the loop annotations of all reference panel samples

(allowing 2-bin mismatches) and kept only annotations that (i) were present in at least 5 reference samples, but (ii) were absent (Chromosight score less than 0) in GM12878, obtaining 170,283 negative pairs. Overall, the entire set contains 74,855 unique positive and 256,609 unique negative examples.

For TAD boundary annotations, we first applied RobusTAD on the combined GM12878 Hi-C contact map and reference samples to obtain TAD boundary scores and identified boundaries. By merging TAD boundaries that were identified from all samples while excluding duplicates (allowing a 2-bin shift), we collected 48,945 loci. We then selected another 54,464 loci by picking one locus every five bins along every autosome. We define the targets for the 103,409 examples as RobusTAD scores from the combined GM12878 Hi-C data. In addition, we created another 103,409 examples at the same loci by using features from a shuffled GM12878 Hi-C contact map and the corresponding RobusTAD scores as targets. In total, there are 206,818 examples.

## Model training and evaluation

The model was trained, evaluated, and tested on contact maps downsampled from the combined GM12878 Hi-C data. During model development, we used chr11 and chr12 for validation, chromosome 15–17 for testing, and the rest of autosomes for training. For loop prediction, the dataset contains 260,940 training, 34,174 validation, and 36,350 test examples. For TAD prediction, the dataset contains 164,458 training, 22,449 validation, and 19,911 test examples. RefHiC takes feature vectors from the study and reference samples as input in the forward pass. To reduce training computation, we sampled 10 reference samples for each example in each epoch independently. During evaluation, we used all samples in the reference panel. For both TAD and loop models, we trained models with a batch size of 1024 for 1000 epochs on an RTX6000 GPU and used AdamW optimizer[48] (weight_decay = 0.1; learning rate = 1e−3). We selected the learning rate that yields the highest validation accuracy in our grid search and used early stopping to prevent overfitting. In the first 5 training epochs, we warmed up the learning rate from 0 to the initial learning rate (i.e., 1e−3) and then reduced the learning rate to 1e−6 in the first 95% epochs using the cosine annealing learning rate scheduler. In addition, we used dropout (rate = 0.25), batch normalization, and layer normalization to regularize network training. We trained the TAD boundary model with MSE loss, and the loop model with focal loss $-(1 - p_t)^\gamma \log(p_t)$ $(\gamma = 2)$[49]. To handle various sequencing depths in a single model, which many existing machine learning applications in Hi-C data analysis are unable to do[43,44], we performed data augmentation by downsampling Hi-C contact maps during training. This transformation preserves topological structures in Hi-C data. However, a Hi-C contact map is too large, and downsampling on the fly is infeasible. We downsampled Hi-C training data and stored them on disk in advance. During training, we randomly selected one contact map from these downsampled contact maps for each training example in each epoch independently. This operation seamlessly worked as a data augmentation by downsampling Hi-C contact map operator during training.

## Contrastive pretraining

We pre-trained the encoder by supervised contrastive learning[42] using Hi-C contact maps downsampled from the combined GM12878 Hi-C data. For each training example, we defined items extracted from the downsampled contact maps at the sample locus as similar items and all Hi-C contact map submatrices in the same batch with different labels as negative items. We aimed to train the encoder such that the distances of embeddings for a training example and its similar items are as close as possible while of embeddings between a training example and its negative items are as far as possible. Following ref. 42, we defined the loss for training instance

$i$ as cross-entropy with in-batch negatives

$$l_i = -\log \frac{e^{\text{sim}(\mathbf{h}_i, \mathbf{h}_i^+)/\tau}}{\sum_{j \neq i} e^{\text{sim}(\mathbf{h}_i, \mathbf{h}_j^-)/\tau}} \qquad (2)$$

where $\mathbf{h}_i$, $\mathbf{h}_i^+$, and $\mathbf{h}_j^-$ are embeddings: $\mathbf{h}_i$ represents item $i$, $\mathbf{h}_i^+$ represents one of item $i$'s similar items, $\mathbf{h}_j^-$ represents an item with a label different from $i$ (i.e., negative item). $\tau$ is a temperature that controls training, and we set it as 1. We pre-trained the encoder for 20 epochs with the LARS algorithm[50] using Adam as a base optimizer. We set batch size to 512 and learning rate to 0.1 during training.

## Hi-C data downsampling

We downloaded the combined Hi-C contact map (.mcool file) for GM12878 cells from 4DN Data Portal (https://data.4dnucleome.org). We downsampled the combined Hi-C contact map to train RefHiC and evaluate sequencing depths' impact on annotating topological structures. We did bilinear downsampling with the downsample function provided in FAN-C[51] from the combined Hi-C contact map to get a series of downsampled data until reaching at ~62M valid read pairs.

## Loop detection with Chromosight, Peakachu, Mustache, and HiCCUPS

We used a variety of loop prediction tools to benchmark against RefHiC. They are executed as follows. Chromosight: We applied the program to each Hi-C contact map with parameters 'detect -p 0.2' to detect loops, sorted detected loops according to scores, and selected the top loops from our test chromosomes. Peakachu: We trained different models for different sequencing depths on GM12878 Hi-C data using our training and validation examples. To match RefHiC, we set the width parameter to 10 and other parameters as default values. We applied the trained models to Hi-C contact maps, adjusted the probability threshold in its pool function to identify loops, sorted loop annotations, and included top loops from test chromosomes as its predictions. Mustache: We used the program by adjusting '-pt' and '-st' to detect at least 1700 loops on our test chromosomes, sorted and selected top loops according to FDR. HiCCUPS: We converted .mcool to .hic files at 5 kb resolution using the 'pre' function provided in Juicer[52]. We applied HiCCUPS by adjusting the '-f' parameter to detect at least 1700 loops on our test chromosomes, sorted and selected top loops according to FDR (obtained as the product of FDR for different filters) as HiCCUPS' prediction. To evaluate the performance of the recommended setting of each tool, we also applied them to annotate loops with their recommended parameters and set FDR as 5% whenever possible.

## TAD detection with alternative tools

We used a variety of TAD callers to benchmark against RefHiC. All tools take .mcool file or files coverted from .mcool file as input. We ran TopDom, Armatus, Arrowhead, EAST, CaTCH, Domaincall (DI), GMAP, ICFinder, and HiCSeg as suggested in ref. 8. We have updated parameters to reflect that we were analyzing data at 5 kb resolution, as needed. We ran HiTAD and deDoc with their default settings. OnTAD: We set maxsz = 600 to allow OnTAD to detect TADs as large as 3 Mb. Grinch: following Lee and Roy[18], we detected TADs by setting the expected TAD length as 2 Mb, 1 Mb, and 500 kb in three runs and combined all results. Supplementary Table 4 contains parameters that we used to execute each TAD caller.

We also compared RefHiC's boundary prediction to two boundary prediction tools. We reimplemented RobusTAD in Python (https://github.com/zhyanlin/RobusTAD)[53] and used Insulation score (IS) function in cooltools[54] in our study. Both take a .mcool file as input. We used RobusTAD to calculate TAD boundary scores and identified

boundaries with default parameters. We ran IS with win = 10 to detect TAD boundaries. As IS only detected insulating bins, to assign boundary orientation (i.e., left vs right), we used RobusTAD's left and right boundary scores to classify IS annotations.

## Enrichment analysis of structural proteins and Histone-3 marks at predicted TADs

To compare the performance of TAD callers, we used an established TAD caller benchmarking scripts[8] to study Histone-3 marks and structural proteins enrichment inside TADs or at TAD boundaries. Briefly, we downloaded ChIP-Seq peak files for CTCF (ENCFF796WRU), RAD21 (ENCFF662DRZ), SMC3 (ENCFF887CRE), H3K36me3 (ENCFF171MDW), and H3K27me3 (ENCFF039JOT) from ENCODE[26]. For structural protein enrichment, we counted the average number of peaks per 5-kb intervals within the regions flanking predicted TAD boundaries (±500 kb). Next, we computed the fold-change as the average peaks in a narrow interval surrounding a boundary (±10 kb) over the average peaks coverage at distant flanks (±400–500 kb). For Histone-3 marks enrichment analysis, we split TADs into 20 kb intervals, summed ChIP-Seq signals inside each interval, computed the log10-ratio of H3K27me3 and H3K36me3 signals (LR), and obtained the average LR for each TAD. We then shuffled the LR values ten times to compute an empirical $p$-value for within-TAD LRs and corrected the $p$-value with the Benjamini–Hochberg procedure to select TADs with significant preference for high or low ratios (FDR ≤ 0.1). To compare TAD partitions, following Zufferey et al.[8], we used the Measure of Concordance (MoC), which ranges from 0 (absence of concordance) to 1 (full concordance) and is defined as follows,

$$\text{MoC}\,(\mathbf{P},\mathbf{Q}) = \begin{cases} 1 & \text{if } N_P = N_Q = 1 \\ \frac{1}{\sqrt{N_P N_Q - 1}}\left(\sum_{i=1}^{N_P}\sum_{j=1}^{N_Q}\frac{|\mathbf{F}_{i,j}|^2}{|\mathbf{P}_i||\mathbf{Q}_j|} - 1\right) & \text{otherwise} \end{cases} \quad (3)$$

where $\mathbf{P} = \{\mathbf{P}_i\}$, and $\mathbf{Q} = \{\mathbf{Q}_j\}$ are sets of TADs including $N_P$ and $N_Q$ TADs, $\mathbf{F}_{i,j}$ is the overlap region between $\mathbf{P}_i$ and $\mathbf{Q}_j$, and $|\cdot|$ represents cardinality. MoC does not handle nested TADs, thus we only included TADs without any smaller TAD in this analysis.

## Enrichment analysis of transcription factors and histone modifications at loop anchors

We downloaded ENCODE ChIP-Seq peak files for 122 TFs and 11 histone modifications in the GM12878 from the UCSC genome browser[26,55] and calculated occupancy fold changes for each TF at loop anchors. We first created a list of unique loop anchors inferred by each tool. For each TF, we counted the number of anchors that overlapped with at least one binding site. We denoted this value as the target. For each chromosome, we randomly created 100 control sets of anchors from the whole genome excluding blacklisted regions[26]. The number of anchors in each control set equals the number of loop anchors in the target set. We then computed the expected overlaps as the mean of overlaps between each control set and the TF's binding sites. Last, we computed fold change as the ratio between the target and the expectation calculated based on control sets.

## Hi-C reference panel

Human reference panel: We downloaded Hi-C sequencing data from the GEO repository and processed them with distiller (https://github.com/open2c/distiller-nf). Briefly, we used bwa mem[56] to map reads to hg38 with option '-SP' and processed the aligned reads with pairtools (https://github.com/open2c/pairtools) to remove duplicates and low-quality read pairs (MAPQ < 10). We then created and normalized contact matrices at 5 kb resolutions using cooler[57] and saved contact maps in .mcool files. Last, we converted these .mcool files into the .bcool file format using cool2bcool function provided in RefHiC. The .bcool

format represents a Hi-C contact map as a band matrix and enables fast random access to square submatrices. Supplementary Table 1 lists all Hi-C data sets included in the human reference panel. Mouse reference panel: We downloaded 20 Hi-C contact maps from 4DN Data Portal (https://data.4dnucleome.org) and processed them as for human. Supplementary Table 2 lists all Hi-C data sets included in the mouse reference panel. Our distributed reference panels contain the aforementioned reference samples. In our experiments, we excluded samples that belong to the study sample's cell type from the reference panel to prevent potential data leakage.

## RefHiC implementation

RefHiC is a Python program available at https://github.com/BlanchetteLab/RefHiC. We implemented the neural network with the PyTorch library[58], and the filtering components for TAD and loop selection with libraries including Pandas[59], SciPy, and NumPy[60]. Using RefHiC to predict loops or TAD boundary scores requires loading data from the study and reference Hi-C contact maps. To reduce memory usage, we extended the Cooler[57] library by implementing a band matrix representation for a contact map and a square function to fetch contact pairs in a given square region and used it to read Hi-C contact maps. ReHiC can make predictions on both CPU and GPU, but is much faster on the latter. RefHiC requires at least 3GB free space for saving reference panel data and at least 12GB RAM for loading reference samples during prediction. We tested RefHiC to annotate TAD boundaries and loops using 20 CPU threads and an RTX6000 GPU. RefHiC calculated TAD boundary scores for whole genome annotation at 5 kb resolution in 30 min. It is impractical and unnecessary to calculate loop scores for all pairs of loci. RefHiC only computes loop scores at bin pairs located within 3 Mb and for which at least one read pair is observed. Thus, the loop annotation running time depends on the study contact map. For instance, it annotates Hi-C data containing 500M valid read pairs in 275 min and Hi-C data containing 250M valid read pairs in 180 min.

## Reporting summary

Further information on research design is available in the Nature Portfolio Reporting Summary linked to this article.

## Data availability

The data that support this study are available from the corresponding author upon reasonable request. All data used in this study are publicly available and their reference numbers are listed in Supplementary Tables 1, 2, and 3. Hi-C contact maps were obtained from 4DN data portal with the following accession code: 4DNFIXP4QG5B (GM12878), 4DNFI4DGNY7J (K562), 4DNFIJTOIGOI (IMR90), 4DNFILP99QJS (HCT-116), and 4DNFIDA2WGV8 (mESC). ChIP-Seq data were obtained from the ENCODE portal with the following accession code: ENCFF796WRU (GM12878 CTCF), ENCFF039JOT (GM12878 H3K27me3), ENCFF662DRZ (GM12878 RAD21), ENCFF171MDW (GM12878 H3K36me3), ENCFF887 CRE (M12878 SMC3), ENCFF508CKL (mESC CTCF), ENCFF203SRF (IMR-90 CTCF), ENCFF119XFJ (K562 CTCF). The CTCF ChIA-PET for IMR-90 were obtained from ENCODE with accession code ENCFF682YFU. The CTCF ChIA-PET for mESC were obtained from ENCODE with accession code ENCFF550QMW. The RAD21 ChIA-PET for GM12878 were obtained from ENCODE with accession code ENCLB784HEF. The CTCF ChIA-PET for K562 were obtained from ENCODE with accession code ENCFF001THV. The RAD21 ChIA-PET for K562 were downloaded from the GEO repository with accession code GSM1436264. The H3k27ac HiChIP data for GM12878 were obtained from ref. 28. The SMC1 HiCHIP data for GM12878 were obtained from ref. 27. The CTCF ChIA-PET data for GM12878 were obtained from ref. 25. Experiment results and intermediate data generated in this study have been deposited in the zenodo repository with https://doi.org/10.5281/zenodo.7133194[61].

## Code availability

Software and documentation available at https://github.com/BlanchetteLab/RefHiC or at this https://doi.org/10.5281/zenodo.7324669[62]. All scripts required to reproduce figures and analyses are available at https://doi.org/10.5281/zenodo.7133194[61].

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

## Acknowledgements

The authors thank Dr. Yue Li, Dr. Jacek Majewski and members of M.B.'s laboratory Audrey Baguette, Zichao Yan, and Elliot Layne for useful discussions in this project, and Audrey Baguette for testing RefHiC. This work was funded by Genome Quebec/Canada and a Genome Quebec/Oncopole/IVADO grants to M.B., and FRQNT Doctoral (B2X) Research Scholarships to Y.Z.

## Author contributions

Y.Z. and M.B. conceived the study and designed models. Y.Z. implemented models, performed data analysis, and wrote the manuscript. M.B. supervised the project and wrote the manuscript. All authors read and approved the final paper.

## Competing interests

The authors declare no competing interests.
