## [Peer Review File · Nature Communications]

REVIEWER COMMENTS

Reviewer #1 (Remarks to the Author):

The paper by Zhang and Blanchette deals with the problem of detecting topologically associating domains (TADs) and loops from the contact maps obtained from Hi-C experiments. The problem is challenging due to the presence of different sources of biases, noise, low coverage, in contact maps. The problem has been studied extensively and several tools exist for detecting TADs and loops. Loop detection is harder than TAD identification, in particular for low coverage data.

Now that many Hi-C contact maps are available, the authors propose to take advantage of these resources by designing an ML framework called RefHiC. The novelty here is to use an attention-based deep learning architecture that was trained from a reference panel of high-quality Hi-C data (e.g., contact maps for GM12878 and corresponding "gold-standard" loops). Algorithmically, the approach uses contrastive pre-training and data augmentation by downsampling.

For the detection of loops RefHiC is compared against ChromSDE, Peakachu, Mustache, and HiCCUPS. RefHiC performs better in all the standard metrics (CTCF, RAD21, SMC1, etc). For TADs RefHiC was compared against RobustTAD and insulation score. While RefHiC is trained on GM12878, the authors show that the model learned is not cell-type specific.

The authors show that RefHiC is notably more robust to low sequencing depth. This robustness, accuracy, and reliability is attributable to the use of reference panel (as explained in Suppl text 2).

The github page is well organized and would allow (at least in principle) to retrain RefHiC for a different species (assuming one had an annotated training set for loops/TADs and access to a GPU).

Concerns:

* The method proposed performs well and is cleverly designed, but it suffers from a major limitation that the authors fail to explicitly address. RefHiC relies on the availability of high quality annotation of loops in order to be trained properly. None of the other competing methods require training, thus they can be applied to species for which a "gold-standard" of long-range loops identified by ChIA-PET or other methods does not exist. The authors show that RefHiC is stable across human cell types, and between human to mouse but the method is likely not going to work on non-mammalian species. The training set gives RefHiC the competitive advantage, but the price paid is that it is not going to be generally applicable across the entire spectrum of species.

* The experimental results on TADs are a bit underwhelming due to the limited number of tools compared. The authors decided to compare RefHiC only against RobustTAD (which they reimplemented) and insulation score. A few more

up-to-date tools needs to be included for the comparison to be convincing. For instance in the paper <https://doi.org/10.1038/s41467-018-05691-7> the authors compares their method deDoc against Armatus, TADtree, Arrowhead, MrTADFinder, Domaincall and CNM.

* I am confused by the fact that the training set for TADs was build using RobusTAD, but when RefHiC is compared against RobusTAD, RefHiC preforms better that RobusTAD; it does not make a lot of sense

Reviewer #2 (Remarks to the Author):

The authors present RefHiC, a tool for identifying loops and topological domains from Hi-C data. The main contribution of this approach is that RefHiC leverages a panel of reference Hi-C data sets from other cell types. Briefly the method works as follows. It takes as input a target Hi-C contact matrix centered on the locus of interest, as well as the corresponding Hi-C matrix in each reference data set, and converts each to a low-dimensional embedding. A attention layer summarizes the reference inputs into a single low-dimensional embedding, which is concatenated with the target embedding. This concatenated embedding is fed into task-specific heads, which output loop and TAD predictions respectively. For loop prediction, following previous work, the network is optimized to recapitulate TF-mediated contacts measured by ChIA-PET or HiCHIP.

The task of identifying loops and TADs is important, and the idea of incorporating reference data sets makes a lot of sense, especially given the expense of generating high-resolution Hi-C data and its consistency across cell types. The manuscript is well-written and the results presented are very good, showing notable improvement over existing methods. However, as described below, I think the paper is missing a key experiment regarding distant cell types that, to me, is the most important question raised by this work. I also have a few concerns about the evaluations, described below.

Comments:

It seems to me that the main question raised by this paper is how distant a target cell type can be while still gaining useful information from the reference Hi-C data sets. When the target cell type is very similar to at least one high-depth reference data set, it seems clear that reference data would help; however, that is not very useful since much of the interesting biology will have already been learned from the reference data. In contrast, the most interesting biology will be learned by profiling novel cell types. The question of whether RefHiC works on a novel cell type seems unanswered in this paper. The authors applied RefHiC to multiple cell types but included all data sets in the reference set (except for data from the target cell type itself). Assaying a novel target cell type could be simulated by removing all similar data sets from the reference, such as all cell types from the same developmental lineage.

Did the authors re-train the alternative methods on the same data sets? I couldn't tell from the text. Particular for Peakachu, it seems that a performance difference could result simply from a different positive/negative set used for training.

Why were the TAD predictions compared only to RobusTAD and insulation score? A 2017 assessment of TAD callers from the authors identified TADtree and TopDom as top performers; a similar 2018 assessment identified TopDom, HiCseg, and CaTCH. Also, these

papers used enrichment of CTCF at TAD boundaries as the primary biological validation. The authors' evaluation was using CTCF ChIA-PET; it is not obvious to me if this is a better or worse evaluation, but CTCF enrichment should be included for consistency with the literature.

Marie Zufferey, Daniele Tavernari, Elisa Oricchio, and Giovanni Ciriello. Comparison of computational methods for the identification of topologically associating domains. *Genome Biology*, 19(1):1–18, 2018.

I must have missed it, but I can't find where the authors described what they used as gold-standard TADs for training the TAD-predictor head.

It would be useful to have a baseline that uses only the reference data. For example, the average prediction across the 5 reference data sets most similar to the target.

Minor notes:

- **The authors should weaken language to avoid asserting that ChIA-PET/HiCHIP represent "true", "actual" or "experimentally-supported" loops. Loops from RefHiC (and HiCCUPS etc) are experimentally-supported, too; they are supported by Hi-C data.**
- **Fig1: I think this would make more sense if there were two head nodes, each pointing to their respective output.**
- **Fig 2i: I can't make out the colors.**
- **Fig2c: Erroneous vertical lines on right side of plot.**
- **Fig 5: "0 sample" > "0 samples"**
- **L358: "For loop annotation, Following"**
- **L390: "the rest autosomes"**
- **L475: "Briefly, We"**

**Maxwell Libbrecht
Assistant Professor
School of Computing Science, Simon Fraser University**

Reviewer #1 (Remarks to the Author):

The paper by Zhang and Blanchette deals with the problem of detecting topologically associating domains (TADs) and loops from the contact maps obtained from Hi-C experiments. The problem is challenging due to the presence of different sources of biases, noise, low coverage, in contact maps. The problem has been studied extensively and several tools exist for detecting TADs and loops. Loop detection is harder than TAD identification, in particular for low coverage data.

Now that many Hi-C contact maps are available, the authors propose to take advantage of these resources by designing an ML framework called RefHiC. The novelty here is to use an attention-based deep learning architecture that was trained from a reference panel of high-quality Hi-C data (e.g., contact maps for GM12878 and corresponding "gold-standard" loops). Algorithmically, the approach uses contrastive pre-training and data augmentation by downsampling.

For the detection of loops RefHiC is compared against ChromSight, Peakachu, Mustache, and HiCCUPS. RefHiC performs better in all the standard metrics (CTCF, RAD21, SMC1, etc). For TADs RefHiC was compared against RobustTAD and insulation score. While RefHiC is trained on GM12878, the authors show that the model learned is not cell-type specific.

The authors show that RefHiC is notably more robust to low sequencing depth. This robustness, accuracy, and reliability is attributable to the use of reference panel (as explained in Suppl text 2).

The github page is well organized and would allow (at least in principle) to retrain RefHiC for a different species (assuming one had an annotated training set for loops/TADs and access to a GPU).

⇒ We thank the reviewer for the overall appreciation of our work.

Concerns:

* The method proposed performs well and is cleverly designed, but it suffers from a major limitation that the authors fail to explicitly address. RefHiC relies on the availability of high quality annotation of loops in order to be trained properly. None of the other competing methods require training, thus they can be applied to species for which a "gold-standard" of long-range loops identified by ChIA-PET or other methods does not exist. The authors show that RefHiC is stable across human cell types, and between human to mouse but the method is likely not going to work on non-mammalian species. The training set gives RefHiC the competitive advantage, but the price paid is that it is not going to be generally applicable across the entire spectrum of species.

⇒ Thank you for your comment. Most Hi-C experiments focusing on 3D genome study are using human or mouse cells. Thus, we anticipate that the majority of RefHiC's potential users will use it to analyze data from those species, and we focused on analyzing human data when developing RefHiC. We agree that it might be more challenging to apply RefHiC to Hi-C data from non-human/mouse species, compared to alternative tools. The primary challenge in applying RefHiC to other species is the lack of Hi-C samples to form a reference panel. However, we argue that RefHiC can work as well as or even better than other tools on a wide range of species given enough Hi-C samples as a reference panel. We agree that RefHiC relies on the availability of high-quality annotation of loops for training, but once trained, we can apply it to other species (e.g. mouse), *without retraining*. In situations where a user thinks the retraining of RefHiC is necessary (e.g. to apply RefHiC to GAM data instead of HiC data) and the user lacks loops identified by ChIA-PET as a "gold standard", the user could still train a loop predictor using high-confidence loop predictions identified by a combination of other loop annotation tools (e.g. Mustache or ChromSight). This training strategy can yield models that outperform these conventional tools, as explained below in our reply to why RefHiC outperforms RobustTAD.

We don't fully agree with the statement that "*None of the other competing methods require training, thus they can be applied to species for which a "gold-standard" of long-range loops identified by ChIA-PET or other methods does not exist.*". Peakachu requires a "gold-standard" of long-range loops as the training target.

ChromoSight does not require training from data, but it requires a predefined loop template (a.k.a kernel) to calculate loop score. This template is a blob-shaped kernel and that was created from yeast *S. Cerevisiae* Hi-C data by the authors, which implies the ChromoSight's kernel is learned from yeast Hi-C data. Mustache is similar. Finally, HiCCUPS scans for locally enriched significant contact pairs following predefined filtering criteria, which were selected by the authors to work well on mammalian HiC data. Hence, although ChromoSight, Mustache, and HiCCUPS are not learning algorithms, they do utilize knowledge learned by human experts on specific data sets (e.g. loops are blob-shaped). Importantly, we only need to train RefHiC and Peakachu once to let them acquire this knowledge. In conclusion, we believe once provided the required input data (i.e. Hi-C contact map, reference panel, etc), all of these blob-shaped pattern detection tools (including RefHiC) can be applied to detect loops in HiC data from new species without retraining.

* The experimental results on TADs are a bit underwhelming due to the limited number of tools compared. The author decided to compare RefHiC only against RobustTAD (which they reimplemented) and insulation score. A few more up-to-date tools needs to be included for the comparison to be convincing. For instance in the paper <https://doi.org/10.1038/s41467-018-05691-7> the authors compares their method deDoc against Armatus, TADtree, Arrowhead, MrTADFinder, Domaincall and CNM.

⇒ Thank you for your suggestion. We have added deDoc, Armatus, Arrowhead, Domaincall and 9 other TAD callers for comparison in the "*RefHiC accurately detects TAD*" section. We tried to compare RefHiC with MrTADFinder, CNM, and TADtree but failed to run them. MrTADFinder does not provide a sample input file and it crashed on our data due to runtime errors. CNM program is not publicly available. TADtree did not finish its first step after running for 5 days (it has three steps), and we terminated it.

* I am confused by the fact that the training set for TADs was build using RobustTAD, but when RefHiC is compared against RobustTAD, RefHiC preforms better that RobustTAD; it does not make a lot of sense

⇒ During training, we used as targets the TAD boundary scores calculated by RobustTAD from the combined GM12878 Hi-C contact map (containing 4B valid read pairs). Our contrastive pretraining and data augmentation by downsampling the Hi-C contact map techniques help RefHiC learn to predict the target from Hi-C contact maps with lower sequencing depths. It is on those lower-depth contact maps that RefHiC can outperform RobustTAD.

Reviewer #2 (Remarks to the Author):

The authors present RefHiC, a tool for identifying loops and topological domains from Hi-C data. The main contribution of this approach is that RefHiC leverages a panel of reference Hi-C data sets from other cell types. Briefly the method works as follows. It takes as input a target Hi-C contact matrix centered on the locus of interest, as well as the corresponding Hi-C matrix in each reference data set, and converts each to a low-dimensional embedding. A attention layer summarizes the reference inputs into a single low-dimensional embedding, which is concatenated with the target embedding. This concatenated embedding is fed into task-specific heads, which output loop and TAD predictions respectively. For loop prediction, following previous work, the network is optimized to recapitulate TF-mediated contacts measured by ChIA-PET or HiCHIP.

The task of identifying loops and TADs is important, and the idea of incorporating reference data sets makes a lot of sense, especially given the expense of generating high-resolution Hi-C data and its consistency across cell types. The manuscript is well-written and the results presented are very good, showing notable improvement over existing methods. However, as described below, I think the paper is missing a key experiment regarding distant cell types that, to me, is the most important question raised by this work. I also have a few concerns about the evaluations, described below.

⇒ We thank the reviewer for their overall appreciation of our work.

Comments:

It seems to me that the main question raised by this paper is how distant a target cell type can be while still gaining useful information from the reference Hi-C data sets. When the target cell type is very similar to at least one high-depth reference data set, it seems clear that reference data would help; however, that is not very useful since much of the interesting biology will have already been learned from the reference data. In contrast, the most interesting biology will be learned by profiling novel cell types. The question of whether RefHiC works on a novel cell type seems unanswered in this paper. The authors applied RefHiC to multiple cell types but included all data sets in the reference set (except for data from the target cell type itself). Assaying a novel target cell type could be simulated by removing all similar data sets from the reference, such as all cell types from the same developmental lineage.

⇒ Thank you for your comment. This is a great point. In our original version, we tried to address this concern by demonstrating that (1) RefHiC can predict loops of different prevalence in the reference panel (Fig. 5 in the manuscript), and (2) RefHiC does not introduce more false positives than alternative tools on cohesin-depleted data (Fig. 3a in the manuscript). We agree that more experiments are needed to evaluate RefHiC's ability to handle novel cells adequately. We have now added two additional experiments:

1. When conducting loop prediction for our test chromosomes (chromosome 15-17), we used reference panel data coming from a different chromosome (chr2). For instance, when computing the loop score for the pair (chr15:100000-105000, chr15:180000-185000), we extracted data for (chr2:100000-105000, chr2:180000-185000) from the reference panel. In this case, all loops in our study sample are "novel".
2. As suggested by the reviewer, we used a new reference panel only containing samples that do not come from the same developmental lineage as the study sample. In addition, we excluded from the reference panel samples whose similarity to the study sample exceeded 0.7 (calculated with HiCRep.py (Lin et al.)). The new reference panel contains 8 samples.

We have provided experiment details in Supplementary text 4 and Supplementary Fig. 14.

Did the authors re-train the alternative methods on the same data sets? I couldn't tell from the text. Particular for Peakachu, it seems that a performance difference could result simply from a different positive/negative set used for training.

⇒ Yes, we retrain alternative methods which require training on the same data sets. We described the training of Peakachu on lines 482-487. We also tried Peakachu's official models, and their performance is worse than those trained with our own data sets for two reasons: 1. We set a larger window size (21x21, compared to 10x10 by default). 2. We combined loop targets from different experiments and added more negative cases to the training set. We observed both improved the performance of Peakachu.

Why were the TAD predictions compared only to RobustTAD and insulation score? A 2017 assessment of TAD callers from the authors identified TADtree and TopDom as top performers; a similar 2018 assessment identified TopDom, HiCseg, and CaTCH.

⇒ Thank you for your comment. In our original version, we presented RefHiC as a TAD boundary caller. It did not make full TAD predictions (i.e. pairing of boundaries). Thus, we only compared RefHiC to RobustTAD and Insulation Score.

In this revised version, we extended RefHiC to make full TAD predictions based on TAD boundary predictions. We used approaches similar to TopDom and GMAP to pair left and right boundaries. In this version, we compared RefHiC to 13 TAD callers (including TopDom, HiCSeq, CaTCH) and benchmarked their performance following the approach of Zufferey et al. We tried running TADtree, but it did not terminate after 5 days so it had to be stopped.

Also, these papers used enrichment of CTCF at TAD boundaries as the primary biological validation. The authors' evaluation was using CTCF ChIA-PET; it is not obvious to me if this is a better or worse evaluation, but CTCF enrichment should be included for consistency with the literature.

Marie Zufferey, Daniele Tavernari, Elisa Oricchio, and Giovanni Ciriello. Comparison of computational methods for the identification of topologically associating domains. *Genome Biology*, 19(1):1–18, 2018.

⇒ Thanks for pointing this out. We used CTCF ChIP-seq data for evaluation in our original manuscript, but some of the axis labels of figure 6 were incorrect and this caused this confusion. We have corrected these figures and included many other evaluation metrics as used in (Zufferey et al.) in the manuscript.

I must have missed it, but I can't find where the authors described what they used as gold-standard TADs for training the TAD-predictor head.

⇒ We have separate models for TAD and loop. We used robustTAD's predictions obtained on the combined GM12878 Hi-C data (4B valid read pairs) as gold-standard (lines 420-428 in the manuscript). Despite using robustTAD's prediction as the target, our contrastive pretraining and data augmentation by downsampling the Hi-C contact map training strategy allow us to train a model that outperforms robustTAD, especially on medium and low coverage data.

It would be useful to have a baseline that uses only the reference data. For example, the average prediction across the 5 reference data sets most similar to the target.

⇒ Thank you for your suggestion. We have now added Supplementary Text 5 and Supplementary Fig. 17 for the suggested experiment. Briefly, we used HiCRep.py to evaluate the similarity between our study sample and each sample in the reference panel. We selected the 5 reference data sets most similar to the study sample and applied Mustache and our deep learning baseline to annotate loops for each data. We then identified loops from the reference data with four different approaches and compared the results against RefHiC. RefHiC outperformed all approaches significantly.

Minor notes:

- The authors should weaken language to avoid asserting that ChIA-PET/HiCHIP represent "true", "actual" or "experimentally-supported" loops. Loops from RefHiC (and HiCCUPS etc) are experimentally-supported, too; they are supported by Hi-C data.

⇒ Thank you for your suggestions. We have replaced "true", "actual", and "experimentally-supported" loops with "ChIA-PET/HiCHIP-supported" loops.

- Fig1: I think this would make more sense if there were two head nodes, each pointing to their respective output.

⇒ Thank you for your suggestion. We tried to split the head into two nodes (Figure 1) but we realized it might lead readers to think that a single model handles the two types of annotation, which is not the case. Indeed, we trained separate models for TAD and loop prediction. Thus, we decide to keep our current figure.

Figure 1. Revised overview figure.

- Fig 2i: I can't make out the colors.

⇒ Sorry for our bad design. We have created a new figure using another set of colors and large markers.

- Fig2c: Erroneous vertical lines on right side of plot.

⇒ Thank you for pointing it out. We have updated this figure accordingly.

- Fig 5: "0 sample" > "0 samples"
- L358: "For loop annotation, Following"
- L390: "the rest autosomes"
- L475: "Briefly, We"

⇒ Thank you for pointing these out! We have fixed these typos.

 Maxwell Libbrecht
 Assistant Professor
 School of Computing Science, Simon Fraser University

References

- Bastié, Nathalie, et al. "Smc3 Acetylation, Pds5 and Scc2 Control the Translocase Activity That Establishes Cohesin-Dependent Chromatin Loops." *Nature Structural & Molecular Biology*, vol. 29, no. 6, June 2022, pp. 575–85.
- Lin, Dejun, et al. "HiCRep.py : Fast Comparison of Hi-C Contact Matrices in Python." *Bioinformatics*, Feb. 2021, <https://doi.org/10.1093/bioinformatics/btab097>.

Zufferey, Marie, et al. "Comparison of Computational Methods for the Identification of Topologically Associating Domains." *Genome Biology*, vol. 19, no. 1, Dec. 2018, p. 217.

REVIEWERS' COMMENTS

Reviewer #1 (Remarks to the Author):

I disagree with the statement that "Most Hi-C experiments focusing on 3D genome study are using human or mouse cells". Nevertheless, I still think that the authors should make it clear in the manuscript that "it might be more challenging to apply RefHiC to Hi-C data from non-human/mouse species, compared to alternative tools" (authors' words here). This is an important limitation that should be made clear -- it should not preclude publication in Nature Comm, but users need to be aware that RefHiC might not be easy to use for species that are not human nor mouse.

Other than this, I am satisfied with the additional TAD experiments.

Reviewer #2 (Remarks to the Author):

The authors have fully addressed my concerns. The revised manuscript is clearly written and likely to be impactful.

One small typo:

L235, L237: "detect TAD" > "detect TADs"

Reviewer #1 (Remarks to the Author):

I disagree with the statement that "Most Hi-C experiments focusing on 3D genome study are using human or mouse cells". Nevertheless, I still think that the authors should make it clear in the manuscript that "it might be more challenging to apply RefHiC to Hi-C data from non-human/mouse species, compared to alternative tools" (authors' words here). This is an important limitation that should be made clear -- it should not preclude publication in Nature Comm, but users need to be aware that RefHiC might not be easy to use for species that are not human nor mouse.

Other than this, I am satisfied with the additional TAD experiments.

⇒ We thank the reviewer for the overall appreciation of our work. We have updated our manuscript to clarify this limitation in the discussion section.

Reviewer #2 (Remarks to the Author):

The authors have fully addressed my concerns. The revised manuscript is clearly written and likely to be impactful.

⇒ We thank the reviewer for the overall appreciation of our work.

One small typo:

L235, L237: "detect TAD" > "detect TADs"

⇒ Thank you for pointing these out! We have fixed these typos.